# Transcriptomic and metabolomic analysis of allelopathic responses in *Elymus nutans*

**Huiyun Yu**[1,2], **Xingming Liu**[2], **Jun Yin**[3], **Lianping Yu**[2], **Lijing Yang**[4], **Yifu Wen**[1*],
**Jie Zhang**[2]

**1** Faculty of Animal Science and Technology, Yunnan Agricultural University, Kunming, China,
**2** Grassland Technology Extension Station of Gansu Province, Lanzhou, China, **3** Yunnan Academy of
Forestry and Grassland Sciences, Kunming, China, **4** Agronomy College, Gansu Agricultural University,
Lanzhou, China

* 2001041@ynau.edu.cn

and Science College, INDIA

**Peer Review History:** PLOS recognizes the
benefits of transparency in the peer review
process; therefore, we enable the publication
of all of the content of peer review and
author responses alongside final, published
articles. The editorial history of this article is
available here: https://doi.org/10.1371/journal.
pone.0349098

## Abstract

To explore the allelopathic response mechanism of *Elymus nutans* to the extracts of
*Ligularia sagitta*, seed germination tests combined with integrated transcriptome and
metabolome analyses were performed to systematically investigate the effects of
the extracts at different concentrations on its germination characteristics and molecular metabolism. The results showed that the concentration range of 0.01–0.10 mg/
mL was the sensitive interval for the allelopathic response of *Elymus nutans* to the
extracts of *Ligularia sagitta*, and germination energy (GE) exhibited higher sensitivity to allelopathic stress than germination rate (GR). Transcriptome analysis yielded
a total of 367463 Unigenes. In the comparison group of A and B,5617 differentially
expressed genes (DEGs) were identified, including 3122 down-regulated genes and
2495 up-regulated genes. GO enrichment analysis revealed that these DEGs were
mainly involved in biological processes such as biosynthesis and structural constituents of ribosomes. KEGG analysis indicated that 2683 DEGs were mapped to 133
pathways, of which 14 were significantly enriched, mainly covering pathways related
to secondary metabolite biosynthesis, ribosome, and plant hormone signal transduction. Metabolome analysis identified 361 differential metabolites (DMs), which were
annotated to 79 pathways with 29 significantly enriched ones. Integrated transcriptome and metabolome analysis revealed that DEMs and DMs were mainly enriched
in two KEGG functional categories, namely Genetic Information Processing and
Metabolism. Among these, the phenylpropanoid biosynthesis pathway was identified
as the core secondary metabolic pathway in response to allelopathic stress. This
pathway relies on a series of enzymatic reactions catalyzed by key genes including
phenylalanine ammonia-lyase (*PAL*), phenylalanine/tyrosine ammonia-lyase (*PTAL*),
cinnamyl alcohol dehydrogenase (*CAD*) and cinnamoyl-CoA reductase (*CCR*),
together with key metabolites such as p-Coumaric acid, Ferulic acid and Sinapyl
alcohol, to produce metabolic products including lignin and flavonoids. This study

**Data availability statement:** The raw sequence data reported in this paper have been deposited in the Genome Sequence Archive (Genomics, Proteomics & Bioinformatics 2025) in National Genomics Data Center (Nucleic Acids Res 2025), China National Center for Bioinformation / Beijing Institute of Genomics, Chinese Academy of Sciences (GSA: CRA033148) that are publicly accessible at https://ngdc.cncb.ac.cn/gsa/browse/CRA033148.

**Funding:** This work was supported by the Gansu Provincial Forestry and Grassland Science and Technology Project (Project No. 2022kj074) and the Yunnan Plateau Characteristic Agriculture Science and Technology Program Project (Project No. 202502AE090016).

**Competing interests:** The authors have declared that no competing interests exist.

provides a theoretical basis for revealing the molecular mechanism underlying the allelopathic response of *Elymus nutans* to the extracts of *Ligularia sagitta*, and also offers a scientific basis for the conservation and sustainable utilization of grassland ecosystems in alpine regions.

## Introduction

*Elymus nutans* is a core native grass species in the alpine ecosystem of the Qinghai-Tibet Plateau, boasting multiple values such as ecological restoration, animal husbandry support, water conservation, and carbon sequestration [1]. Its healthy growth is of great significance for regional ecological stability, people's livelihood, and sustainable development. In natural grasslands, the growth of *Elymus nutans* is affected by various factors including climate, soil, and interspecific competition, which further affects the overall ecological stability, production efficiency, and sustainable development of grasslands on the Qinghai-Tibet Plateau. Thus, it serves as a key link in maintaining the regional ecological security barrier, ensuring the foundation of alpine animal husbandry, and promoting the coordinated development of ecological governance and people's livelihood improvement. In natural grasslands, weeds inhibit the growth and development of *Elymus nutans* through plant competition, resource competition, and allelopathy [2], thereby affecting its dominance in the community and the overall productivity of the grassland. Currently, researchers have conducted extensive studies on interspecific competition [3] and resource competition [4] between *Elymus nutans* and other plants. However, the allelopathic effects of weeds on *Elymus nutans* have not received sufficient attention, which to a certain extent restricts the comprehensive understanding of maintaining the healthy growth of *Elymus nutans*.

As a typical allelopathic species in alpine ecosystems and agricultural habitats, the root extract of *Ligularia sagitta* showed inhibition rates of up to 54.54% and 80.33% on the shoot and root length of Poa pratensis [5]. The mechanisms of action involve multiple dimensions, including cell membrane permeability damage, photosynthesis inhibition, as well as abnormal alterations in physiological and biochemical processes and material metabolic pathways [6]. However, obvious limitations still exist in current research, which mostly focuses on the isolation and identification of single active components from *Ligularia* plants [7]. The research content mostly focuses on the allelopathic inhibitory effects of aqueous extracts from its roots, stems, leaves and other organs on seed germination and seedling growth of receptor plants, while the discussion on mechanisms is only limited to photosynthesis interference, cell membrane system damage and other aspects [8,9]. At present, there is still a significant deficiency in the elucidation of molecular regulatory mechanisms underlying the response of receptor plants to allelopathic stress from *Ligularia* plants. In particular, triterpenoids, which are ubiquitous in plants, have been confirmed to exist in the roots of *Ligularia sagitta* and may be involved in the allelopathy process. Nevertheless, their specific action targets, synergistic mechanisms with other terpenoids, and environmental adaptive response patterns remain unexplored research gaps to date.

Behind the physiological responses of plants to allelopathic stress, there are sophisticated regulation of gene expression and dynamic changes in metabolites. The integrated analysis of transcriptome and metabolome has emerged as a powerful tool for elucidating the molecular mechanisms underlying plant stress responses, which can systematically reveal the core regulatory pathways and key metabolites at the transcriptional and metabolic levels [10]. Transcriptome sequencing-based analysis of differentially expressed genes in target plants treated with allelochemicals from Ligularia species enables the precise identification of key regulatory nodes responsible for the suppressed expression of *PAL* and *CHS* genes involved in pathways such as phenylpropane metabolism and flavonoid biosynthesis [11]. By means of untargeted metabolomics technology to capture the dynamic change characteristics of secondary metabolites, we can explore the perturbation rules of terpenoid components on the secondary metabolic network of receptor plants, and then further clarify the intrinsic correlation between the decreased contents of quercetin and ferulic acid and the growth-inhibited phenotype of plants [12]. The research results can break through the limitations of traditional single-component analysis, provide theoretical support for an in-depth understanding of the interactions between allelochemicals and plants, and meanwhile offer a scientific basis for the protection and sustainable utilization of grassland ecosystems in alpine regions.

## Materials and methods

### Experimental materials and treatments

Plants of *Ligularia sagitta* and seeds of *Elymus nutans* were collected from Xiuwa Village, Gahai Town, Luqu County, Gannan Tibetan Autonomous Prefecture (34°13′52″N,102°14′42″E). The collection work has been authorized by the Luqu County Grassland Workstation of Gannan Prefecture. The plant specimens of *Ligularia sagitta* were authenticated by Professor Guo Yehong of Gansu Agricultural University, while those of *Elymus nutans* were identified by Researcher Yang Yandong of the Luqu County Grassland Workstation. All voucher specimens are deposited in Gansu Agricultural University.

The appearance and melting point of the compound were recorded for preliminary type identification. Nuclear Magnetic Resonance (NMR) spectroscopy was used to characterize the molecular skeleton and functional groups, ¹H-NMR was applied to determine the hydrogen environment, and ¹³C-NMR for the carbon skeleton, with tetramethylsilane (TMS) as the internal standard, and chemical shifts expressed as δ (ppm) and coupling constants as J (Hz). Thin-layer chromatography (TLC) analysis was performed under a 254 nm ultraviolet analyzer, using 10% sulfuric acid-ethanol (v/v) as the chromogenic reagent. The chemical structure of the compound was confirmed by comparing the spectral data with standard spectral libraries and literature data.

Seed germination test: The tested grass seeds were plump seeds of *Elymus nutans*, a dominant species in the mountain meadows of Luqu. After sequential treatments of soaking in 60% $H_2SO_4$ for 10 minutes (to corrode the seed coat) → repeated rinsing with distilled water → soaking in 0.2% NaClO for 3 minutes (for sterilization) → rinsing 5–6 times with sterile distilled water, the seeds were evenly placed in 90 × 15 mm Petri dishes lined with two layers of filter paper (40 seeds per dish). Four milliliters of the extract was added to moisten the seeds, and the methanol was volatilized in a fume hood for 1 hour. Seven concentration gradients were set in the experiment (0.2,0.15,0.1,0.05,0.01,0.001 mg/mL, and distilled water treatment as the control check, CK), with 3 replicates per gradient. The Petri dishes were incubated in an artificial climate chamber for 12 days (photoperiod 12 h/12 h, light intensity 10,000 l00d7/dark 0 l00d7, temperature 23 ± 0.5 °C), and distilled water was regularly supplemented during the period to maintain the concentration of the extract. Germination energy (GE) was counted on the 5th day, germination rate (GR) on the 10th day, and 10 seedlings were randomly selected from each gradient on the 14th day to determine their fresh weight, root length, and stem length.

Sample collection for integrated omics analysis. Seedling samples were collected on the 14th day of seedling growth. The groups treated with the compound at concentrations of 0.10 mg/mL and 0.01 mg/mL, as well as the control group, were designated as Group A, Group B and CK, respectively. Three replicate samples were collected from each treatment group, with 9 samples prepared for transcriptome analysis and another 9 for metabolome analysis. All 18 samples were

wrapped in tin foil, labeled with serial numbers, and immediately snap-frozen in liquid nitrogen [13]. All frozen samples were transported to Wuhan Bena Technology Co., Ltd. via a professional cold chain for transcriptome sequencing and metabolomic analysis.

## Experimental method

The calculation formulas for seed germination rate (GR), germination energy (GE), and seedling inhibition rate are presented as follows. Statistical analysis was performed using SPSS 26 software, one-way analysis of variance (ANOVA) was conducted for data analysis, and Duncan's multiple range test was used to determine the significance of differences, with the significance level set at $p < 0.05$.

$$GE = \frac{\text{Number of germinated seeds at peak germination stage}}{\text{Statistics of total seed sample size}} \times 100\%$$

$$GR = \frac{\text{Number of normally germinated seeds within 12 days}}{\text{Statistics of total seed sample size}} \times 100\%$$

$$\text{Inhibition Rate(\%)} = (1 - \frac{\text{Treatment Group Mean}}{\text{Control group mean}}) \times 100\%$$

Transcriptome Sequencing Method. Total RNA was extracted using the Plant RNA Kit. The quality and integrity of the extracted total RNA were detected with a NanoDrop One Spectrophotometer and a Qubit 3.0 Fluorometer. mRNA was enriched with oligo (dT), followed by fragmentation using a fragmentation reagent. First-strand and second-strand cDNA were synthesized, and then the cDNA library was constructed through a series of steps including end repair, sequencing adapter ligation, purification, PCR amplification, and product circularization. The insert size of the library was detected using a Bioanalyzer, and the effective concentration of the library was accurately quantified. After the library passed the quality inspection, sequencing was performed on the Illumina HiSeq platform. DEGs were identified using the thresholds of $p < 0.05$ and $|\log_2 FC| \geq 1$, followed by Gene Ontology (GO) and KEGG enrichment analyses [14,15]. To verify the accuracy of the transcriptome sequencing results, quantitative real-time PCR (qRT-PCR) validation was performed on a total of 8 DEGs.

Metabolomic analysis method. Chromatographic separation of target compounds was performed using a Phenomenex Kinetex C18 liquid chromatography column via the UPLC-MS/MS approach. Primary and secondary mass spectrometry data were collected with an Orbitrap Exploris 120 mass spectrometer. Univariate statistical analysis and multivariate statistical analysis were conducted on the qualitative and quantitative results of the metabolome to screen for metabolites with significant differences. Among these, metabolites meeting the criteria of VIP > 1 and $p < 0.05$ were defined as differential metabolites (DMs).

For data analysis, one-way analysis of variance (ANOVA) was performed on the data using SPSS 26 software, and Duncan's multiple range test was adopted for significance test of differences, with the significance level set at $p < 0.05$.

## Results

### Effects of different concentration treatments on seed germination and seedling growth of *Elymus nutans*

GE exhibited a higher sensitivity to the allelopathic response of MWA than GR, and the concentration range of 0.01– 0.10 mg/mL was the sensitive interval for the allelopathic response of *Elymus nutans* seeds to MWA. When the concentration of MWA was 0.001 mg/mL, there was no significant difference in the GR of *Elymus nutans* seeds compared with

the control group (CK) ($p > 0.05$), whereas GE decreased significantly ($p < 0.05$). At a concentration of 0.01 mg/mL, seed germination was delayed. When the concentration reached 0.1 mg/mL, seed germination was obviously inhibited, with GR and GE reaching 46% and 16.33%, respectively ($p < 0.05$). When the concentration increased to 0.15 mg/mL, seed germination was severely inhibited, and GR and GE further decreased to 17.33% and 6.67%, respectively ($p < 0.05$).(Fig 1)

## Screening and functional enrichment analysis of DEMs

After rigorous quality control and sequence assembly of the sequencing data, a total of 367463 Unigenes were obtained, with an average sequence length of 656 bp. The percentage of Q30 bases in each sequencing library was higher than 93%, and the GC content was 52.06%. All these indicators demonstrated that the sequencing and assembly quality was excellent, which could meet the requirements of subsequent analyses. The results of differential expression analysis showed that a total of 5617 differentially expressed genes were identified in the comparison group of high concentration (A) and low concentration (B), among which 2495 were up-regulated genes and 3122 were down-regulated genes. This comparison group exhibited the expression characteristic that the number of down-regulated genes was greater than that of up-regulated genes (Fig 2).

## GO functional annotation analysis of differentially expressed genes

GO functional annotation and enrichment analyses were performed on the differentially expressed genes, which were then classified according to the three major functional categories: biological process (BP), cellular component (CC) and molecular function (MF). The results showed that a total of 378 BP-related GO terms,311 CC-related GO terms and 113 MF-related GO terms were significantly enriched in the A vs B comparison group. For the A vs B comparison group, in terms of BP, the differentially expressed genes were mainly enriched in biosynthetic process, organic substance biosynthetic process, cellular biosynthetic process, cellular nitrogen compound biosynthetic process and macromolecule biosynthetic process; in terms of CC, they were mainly enriched in intracellular organelle, organelle, non-membrane-bounded organelle, intracellular non-membrane-bounded organelle and ribosome; in terms of MF, they were mainly enriched in structural molecule activity, structural constituent of ribosome, transcription regulator activity, DNA-binding transcription factor activity and tetrapyrrole binding (Fig 3). The enrichment of molecular functions related to structural constituent of

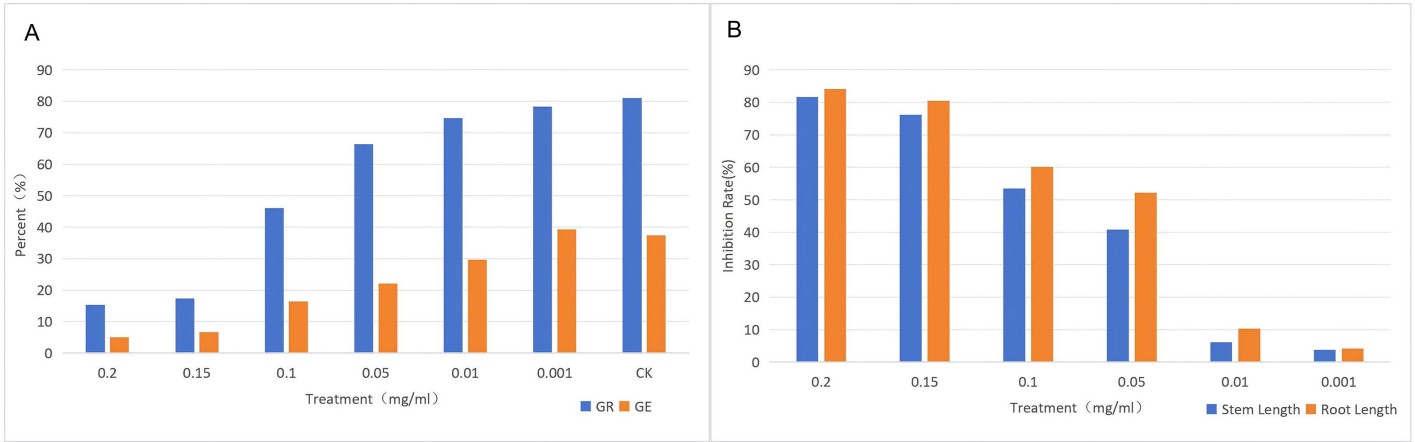

**Fig 1. Effects of different MWA concentrations on GR, GE and inhibition rate of *Elymus nutans* seeds. (A)** Bar chart of the effects of different concentration gradients of MWA on the GR and GE of *Elymus nutans*. **(B)** Bar chart of the inhibition rates of different concentration gradients of MWA on the seedling stem length and root length of *Elymus nutans*.

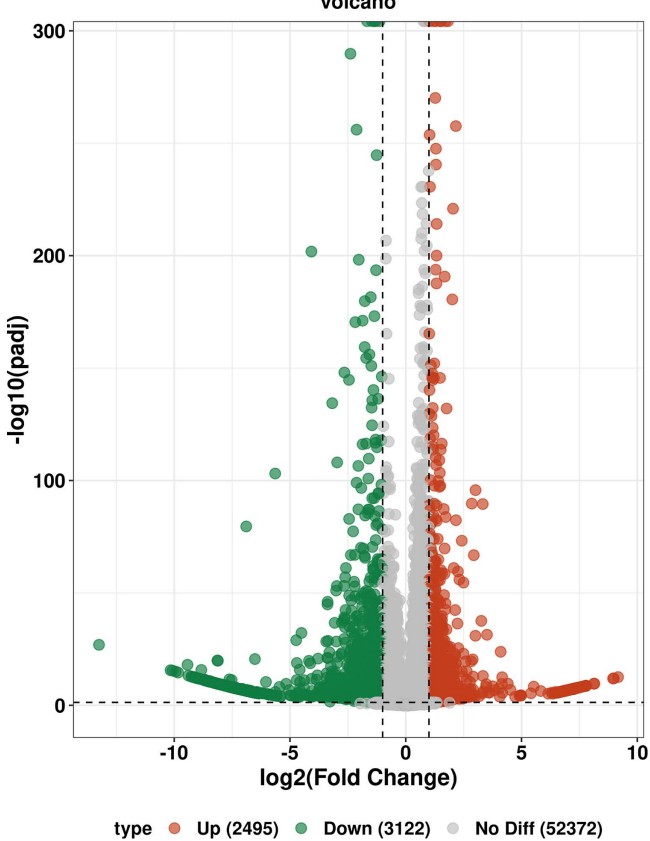

**Fig 2. Volcano plot of DEMs.**

ribosome in MF was consistent with the enrichment of ribosome term in CC classification, which further confirmed that there were significant differences in protein synthesis mechanisms between the two groups.

### KEGG enrichment analysis of DEGs

KEGG pathway enrichment analysis revealed that a total of 2683 DEGs in the A vs B comparison group were mapped to 133 KEGG pathways, with 14 pathways identified as significantly enriched ones (*qvalue* < 0.05). The KEGG pathway classification mainly covers three major categories, namely Metabolism, Genetic Information Processing and Environmental Information Processing. Among the significantly enriched pathways, the A vs B comparison group was mainly enriched in Biosynthesis of secondary metabolites, Ribosome, MAPK signaling pathway-plant, Phenylpropanoid biosynthesis, Plant hormone signal transduction, Biosynthesis of various plant secondary metabolites, Flavonoid biosynthesis, Cutin, suberine and wax biosynthesis, alpha-Linolenic acid metabolism and Stilbenoid, diarylheptanoid and gingerol biosynthesis, which were predominantly up-regulated overall (Fig 4).

### qRT-PCR validation of DEGs

To verify the accuracy of transcriptome sequencing results, 8 DEGs including *C4H*, *KATG*, *PAL*, *PTAL*, *4CL*, *CCR*, *ANR*, and *CAD* were selected for quantitative real-time PCR (qRT-PCR) validation. The sequences of gene-specific primers for each gene are provided in Table 1. The expression pattern change trends of the 8 target

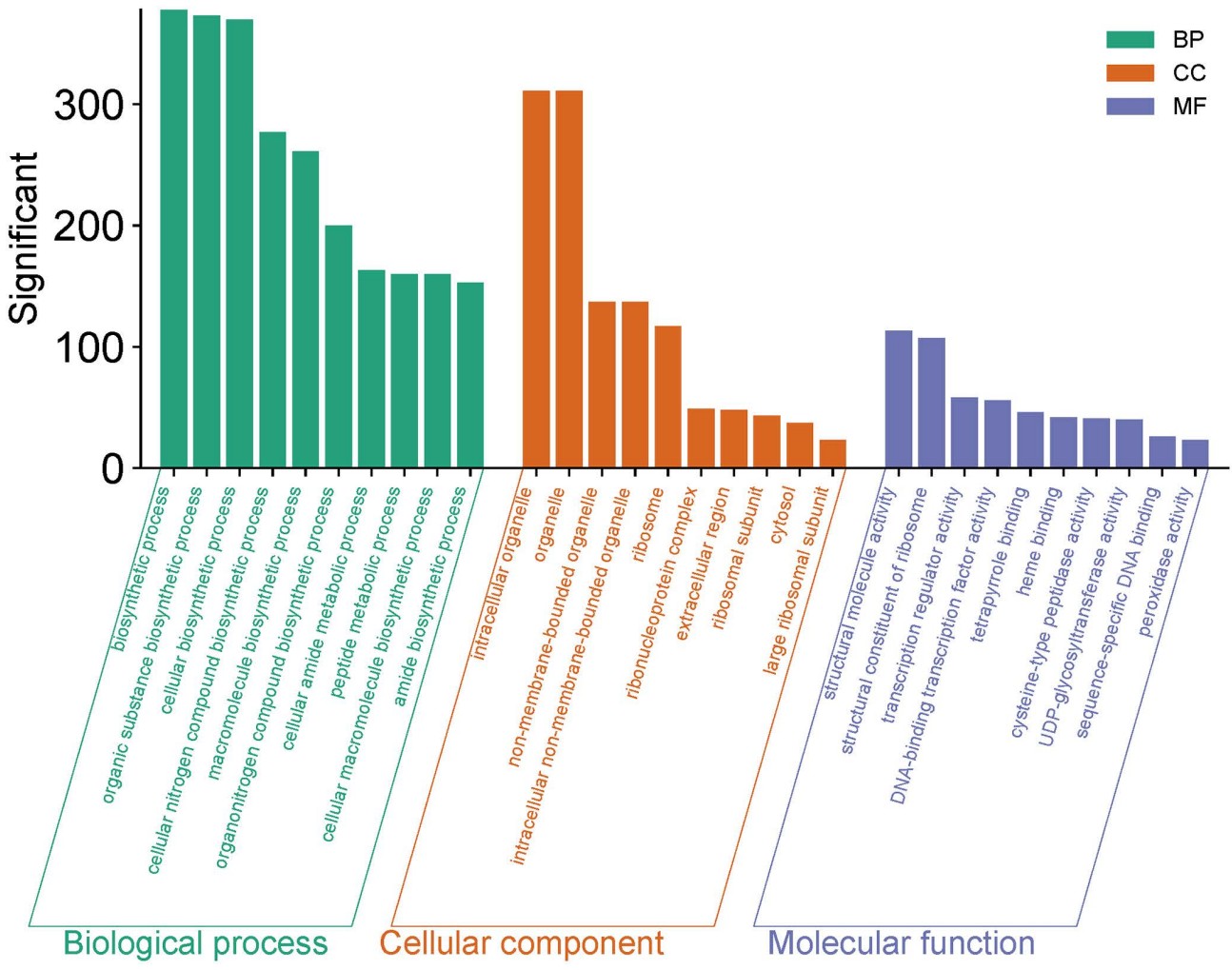

**Fig 3. Combined bar chart of GO term enrichment (BP, CC, MF categories) in 2-A vs 2-B Group.**

genes in Groups A and B were highly consistent with the qRT-PCR validation results, the correlation coefficients ($r$) between the differential expression fold changes ($\log_2$FC) of genes and the validation values in the two groups reached 0.77 and 0.82, respectively (Fig 5). The above results indicated that the transcriptome sequencing data exhibited high reliability, which could accurately reflect the characteristics of differential gene expression in the samples and directly support subsequent studies related to gene function analysis, pathway enrichment and other relevant research.

## Screening and KEGG enrichment of DMs

The quality control (QC) samples showed good clustering in the principal component analysis (PCA) score plot, the correlation coefficients among all samples were greater than 0.95, the relative standard deviation (RSD) of the internal standard response was 0.058, and the median RSD was ≤10%. In addition, all experimental samples were distributed within the 95% confidence interval regardless of the inclusion of QC samples. The above results indicated that the detection data obtained in this study had high reliability.

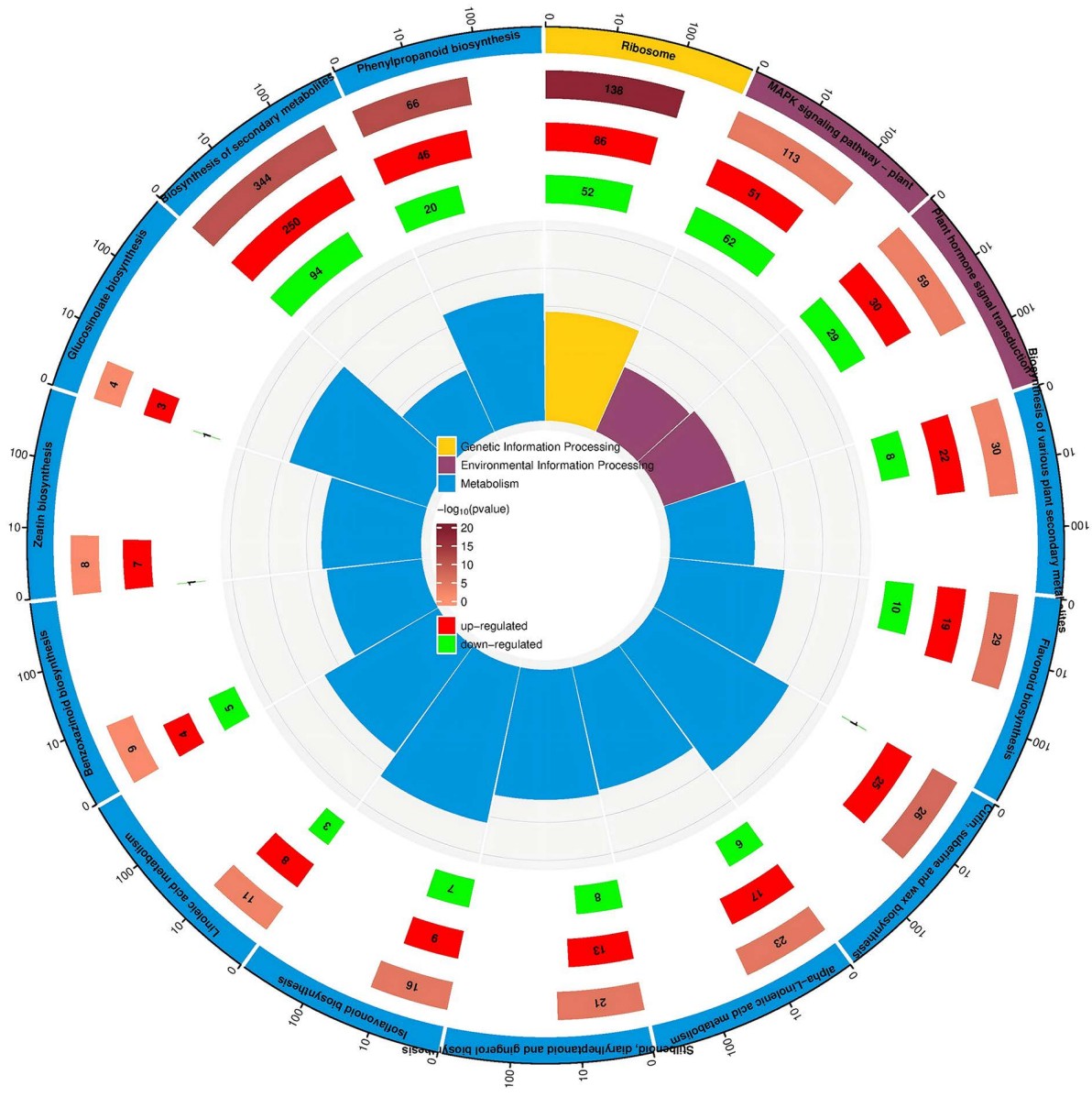

**Fig 4. KEGG enrichment of DEGs in 2-A vs 2-B Group.**

In the A vs B comparison group, a total of 361 DMs were screened out, and these DMs could be annotated to 79 metabolic pathways, among which 29 pathways reached the level of significant enrichment. The top 20 significantly enriched pathways were selected for in-depth analysis, and the results showed that the 361 DMs included 203 up-regulated metabolites and 158 down-regulated metabolites, which were mainly enriched in pathways such as Metabolic pathways, Biosynthesis of secondary metabolites, Biosynthesis of cofactors, ABC transporters and Biosynthesis of amino acids (Fig 6).

**Table 1. Differential gene primer sequence.**

| Gene | Forward primer (5′-3′) | Reverse primer (5′-3′) |
|---|---|---|
| Phenylalanine Ammonia-Lyase (PAL) | TCGCCATGGCCTCGTACTGC | CAGTTCTTGGGCAGCGAGAC |
| Phenylalanine/Tyrosine Ammonia-Lyase (PTAL) | GACGACGCGTACCTCGTCAG | TGGACTATGGTTTCAAGGGCG |
| Cinnamate 4-Hydroxylase (C4H) | TCGACGACGACGAGATCTTC | CGGCTACCTCCGCGGCTACCT |
| 4-Coumarate-CoA Ligase (4CL) | GAGATCTGCATCCGCGGGGAG | GGATCCGCGTATGCGAAGGC |
| Cinnamyl Alcohol Dehydrogenase (CAD) | CGCGTCGACAAGGGGCTCACC | TGAACGACGTGCGCTACCGC |
| Cinnamoyl-CoA Reductase (CCR) | CAGAGATTGTCAAGATCATAC | CTGCTGTAACTTCCCCGCCA |
| Catalase-Peroxidase (katG) | GAGCTTCATCGGGAGCATGG | CTCGTCGACGACGACGAGATC |
| Anthocyanidin Reductase (ANR) | GAGCTTCATCGGGAGCATGG | TGAACGACGTGCGCTACCGC |

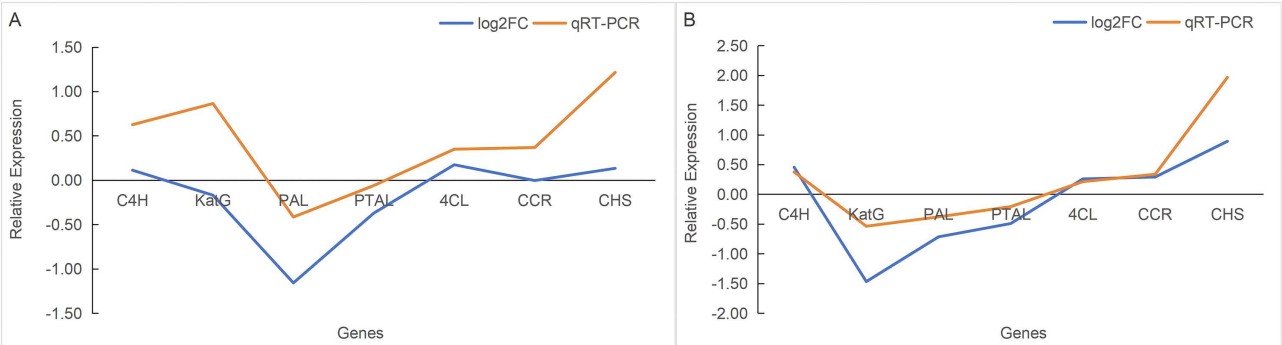

**Fig 5. Linear fitting curve plot of qRT-PCR and RNA-seq.** (A) Trend comparison between transcriptome log2FC and qRT-PCR validation results of key genes in Group A. (B) Trend comparison between transcriptome log2FC and qRT-PCR validation results of key genes in Group B.

## KEGG pathway enrichment and correlation analysis of DEMs and DMs

The results of combined KEGG pathway enrichment analysis of transcriptome and metabolome showed that the DEMs and DMs in the A vs B comparison group were mainly co-enriched in two KEGG functional categories: Genetic Information Processing and Metabolism (Fig 7A). Among these pathways, Ribosome, Phenylpropanoid biosynthesis, Cutin, suberine and wax biosynthesis, Stilbenoid, diarylheptanoid and gingerol biosynthesis, and alpha-Linolenic acid metabolism exhibit high enrichment factors and enrichment intensities, and thus represent the core regulatory pathways of this comparison group (Fig 7B).

Based on the Pearson correlation analysis method, the correlation coefficients between genes and metabolites were calculated, and the correlations between significant DEMs and significant DMs were visualized by a heat map of the correlation coefficient matrix. In the heat map, rows and columns represent the detected substances from transcriptomics and metabolomics, respectively, while the colors indicate different correlation coefficients (Fig 8).

In summary, in the analysis of different concentration treatments, Phenylpropanoid biosynthesis and Ribosome were identified as the common enriched pathways of DEGs and DMs. These two pathways not only represent the core pathways throughout the A vs B comparison group, but also serve as the key regulatory targets in response to stresses of different concentrations, thus occupying the central position in the metabolic regulatory network. Among these pathways, the phenylpropanoid biosynthesis pathway, as a core pathway of plant secondary metabolism, not only governs the biosynthesis of important secondary metabolites such as phenols, flavonoids and lignin, but also participates in a series of key physiological processes including secondary metabolic regulation, defense metabolism and cuticular structure

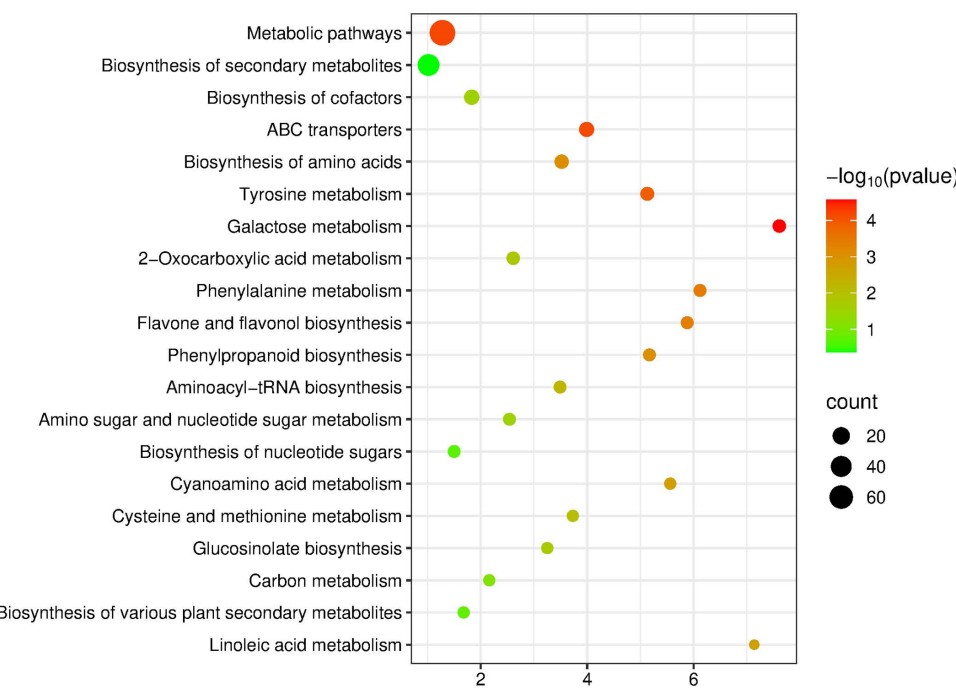

**Fig 6. KEGG enrichment bubble plot of differential metabolites in 2-A vs 2-B Group.**

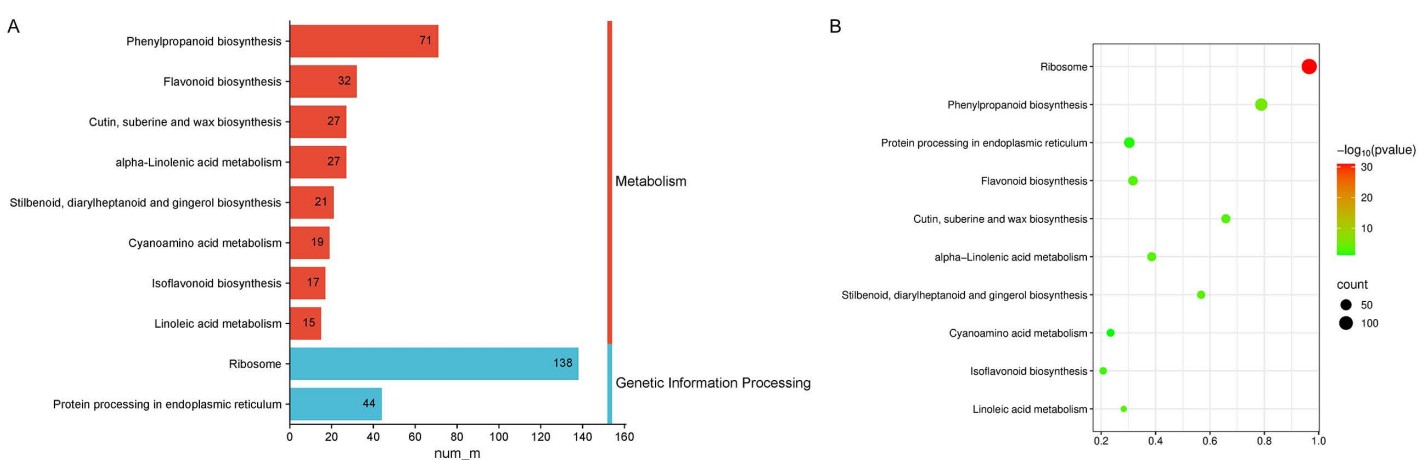

**Fig 7. KEGG pathway enrichment patterns in the 2-A vs 2-B group. (A)** Classification plot of KEGG pathway enrichment results in 2-A vs 2-B Group. **(B)** KEGG pathway enrichment bubble plot in 2-A vs 2-B Group.

biosynthesis, thereby directly affecting metabolite diversity and organismal stress resistance. The ribosome pathway, as the core pathway for protein synthesis, determines the post-transcriptional translation efficiency of genes. It can indirectly regulate this secondary metabolic pathway by modulating the synthesis and accumulation of relevant functional proteins involved in the phenylpropanoid biosynthesis pathway. Based on this, the molecular regulatory mechanism underlying the response of *Elymus nutans* to MWA allelopathic stress can be elucidated with the phenylpropanoid biosynthesis pathway as the core.

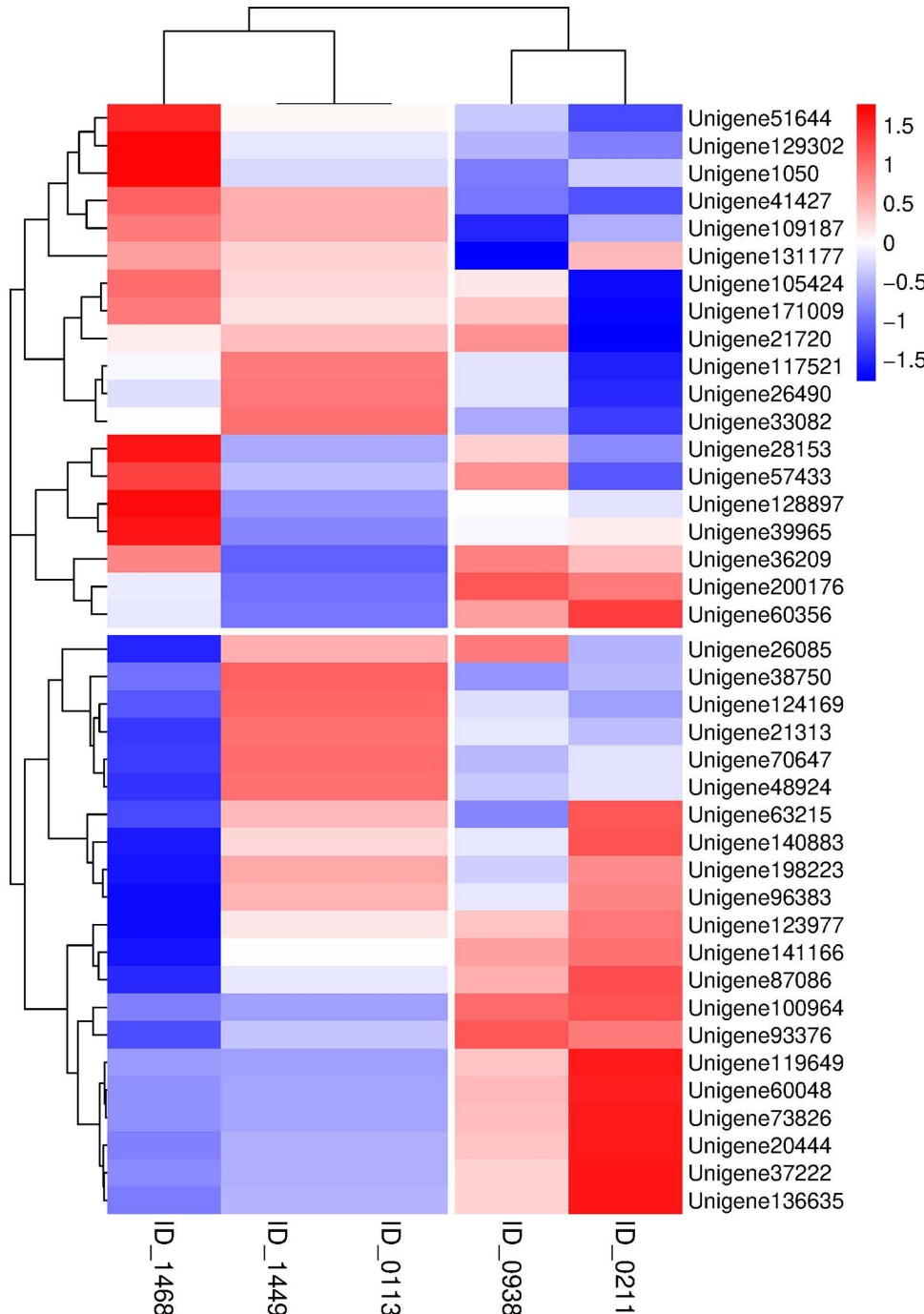

**Fig 8. Heatmap of correlation coefficient matrix between significant DEMs and DMs in 2-A vs 2-B Group.** In the plot, ID_0113, ID_0211, ID_0938, ID_1449 and ID_1468 correspond to L-tyrosine, sinapyl alcohol, ferulic acid,4-hydroxystyrene and p-coumaric acid, respectively.

## Effects on the biosynthesis of phenylpropanoids

The results of the combined analysis indicated that Phenylpropanoid biosynthesis is the core secondary metabolic pathway for *Elymus nutans* seedlings in response to MWA allelopathic stress. This pathway initiates with phenylalanine, which is catalyzed by *PAL* to generate cinnamic acid; the latter is then catalyzed by *C4H* to produce p-hydroxycinnamic acid, followed by the formation of 4-coumaroyl-CoA under the catalysis of *4CL*. Subsequently, the pathway proceeds to the downstream reactions, which convert the intermediate into various phenylpropanoid metabolites, including lignin, flavonoids, and phenols.

The core regulatory mechanism of *Elymus nutans* seedlings in response to stress is as follows: the lignin biosynthesis is multi-dimensionally inhibited by regulating four key stages of the phenylpropanoid pathway, namely the initiation, intermediate branching, precursor synthesis, and terminal polymerization. At the initiation stage, MWA at different concentrations can inhibit the expression of *PAL* and *PTAL*, the rate-limiting enzymes of the pathway initiation, with a more significant inhibitory effect observed under high concentration. As shown in Fig 9, the expression levels of *PAL* in Group A and Group B were −2.25 and −1.79, respectively, while those of *PTAL* were −2.81 and −1.68. The reduced synthesis of these two enzymes leads to decreased production of cinnamic acid and p-coumaric acid, thereby lowering the metabolic flux entering the pathway from the source.

At the intermediate branching regulation stage, the inhibitory isoform of *HCT* (a key enzyme of the pathway branches) dominates the regulation of the lignin branch. Its expression levels were −8.5 and −3.06 in Group A and Group B, respectively, with a weaker inhibitory effect observed under low concentrations than under high concentrations. Specifically, ferulic acid synthesis was increased by 103.88% under low concentrations, whereas it was decreased by 21.40% under high concentrations, the resultant ferulic acid was then converted to sinapic acid via *F5H* and entered the non-lignin branch. Meanwhile, the activating isoform of *HCT* had expression levels of 14.15 and 1.59 in Group A and Group B, respectively, which drives the conversion of p-coumaroyl-CoA to flavonoid/phenolic acid derivatives, reduces the metabolic flux allocation toward lignin synthesis, and thereby enhances stress resistance-related secondary metabolism.

At the precursor synthesis stage, *CAD* (the enzyme required for cinnamyl alcohol synthesis) was downregulated under both high and low concentrations, with its expression levels reaching −7.05 and −3.33 in Group A and Group B, respectively. The inhibitory effect was more pronounced under high concentrations, which directly blocked cinnamyl alcohol production. In contrast, *CCR* was transcriptionally activated under low concentrations with an expression level of 1.25, whereas its compensatory upregulation was insufficient under high concentrations. Consequently, the lignin monomers and their derivatives (sinapyl alcohol and coniferin) were decreased by 99.27% and 46.84% in the high and low concentration groups, respectively, thereby further inhibiting lignin synthesis at the level of raw material supply.

At the terminal polymerization stage, MWA exerted a strong inhibitory effect on peroxidase, with its expression levels being −3.41 and −1.49 in Group A and Group B, respectively. The inhibitory effect under high concentrations was far stronger than that under low concentrations, which directly prevented the polymerization of lignin monomers, downregulated the polymerization intermediate product 5-hydroxystyrene, and ultimately blocked lignin biosynthesis.

## Discussion

### Allelopathic stress effects of MWA on seed germination and seedling growth of *Elymus nutans*

The inhibitory effect of MWA on seed germination and seedling growth of *Elymus nutans* became increasingly significant with the increase in concentration, and there were obvious differences in the response sensitivity of different germination indexes to the stress of this compound. The treatment at the concentration of 0.01 mg/mL only delayed the seed germination process, and no statistically significant differences were observed in the average fresh weight and root length of seedlings compared with the control group. This result was consistent with the relevant rules of allelopathic regulation by *Medicago sativa* extracts reported in previous studies [16]. When the concentration increased to 0.1 mg/mL, seed

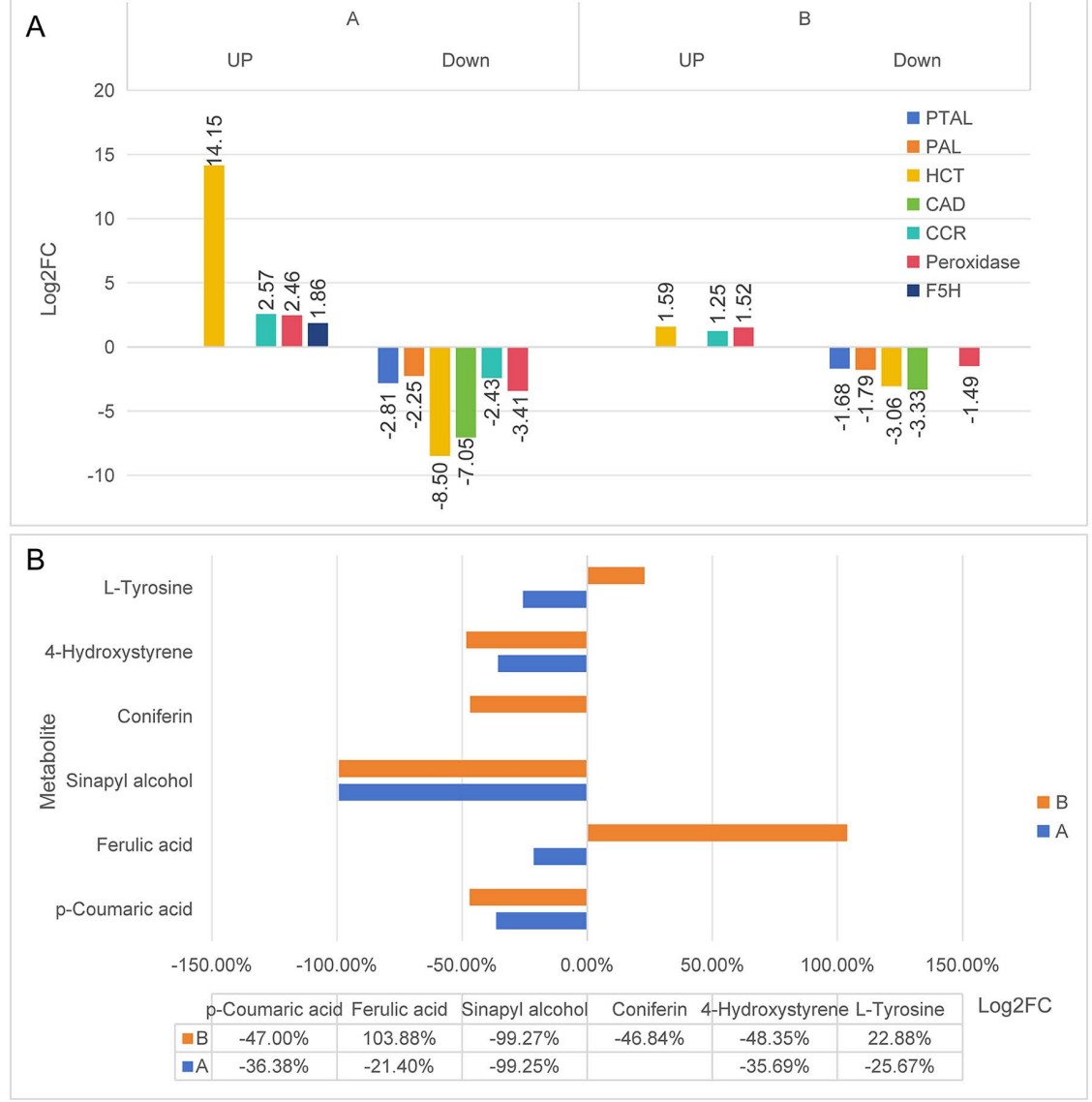

**Fig 9. Phenylpropane biosynthesis pathway expression profile graph. (A)** Relative expression levels of key enzymes (genes) in the phenylpropanoid biosynthesis pathway. **(B)** Relative contents of major metabolites in the phenylpropanoid biosynthesis pathway.

germination was significantly inhibited, with the germination rate (GR) and germination energy (GE) decreasing to 46% and 16.33%, respectively. When the concentration reached 0.15 mg/mL, seed germination was arrested, and the GR and GE were only 17.33% and 6.67%, respectively ($p < 0.05$). The inhibitory effect exhibited an exponentially increasing trend with the rise in concentration, which was similar to the regulatory characteristics of Ageratina adenophora extracts on seed germination of crops such as rice, reflecting the critical inhibitory threshold property of allelochemicals [17,18].

## Expression characteristics and pathway enrichment patterns of DEGs under allelopathic stress

Under high-concentration MWA stress, the number of DEGs in *Elymus nutans* seedlings was significantly higher than that under low-concentration treatment. In all comparison groups, the number of down-regulated genes exceeded that

of up-regulated genes, indicating that high-concentration stress exerted a stronger perturbation effect on plant gene expression.

GO functional annotation revealed that substance anabolism serves as a core process for maintaining cell growth, development, and functional homeostasis. The enrichment of DEGs in these functional terms indicated that there might be significant differences in metabolic activity between Group A and Group B. It was inferred that the treatment differences between A vs B might regulate anabolic pathways, affect the synthesis efficiency of intracellular key substances, and then induce phenotypic differentiation. This viewpoint has been verified by transcriptome analysis studies on barley genotypes under arsenate and phosphate treatments [19]. As the core site for protein synthesis, the significant enrichment of ribosome-related functional terms indicated that there might be differences in protein synthesis capacity between Group A and Group B, while changes in protein synthesis efficiency would directly affect the steady-state levels of intracellular functional proteins [20]. KEGG enrichment analysis indicated that secondary metabolites act as key regulatory substances for plant growth, development and environmental adaptation, and the activation of their biosynthetic pathways is usually closely associated with plant stress responses and phenotypic differentiation [21]. Among these pathways, the phenylpropanoid pathway is the core pathway of plant secondary metabolism, which not only participates in the biosynthesis of important substances such as lignin and flavonoids, but also plays a key role in regulating plant structural stability and stress resistance. As important secondary metabolites, the up-regulation of flavonoid synthesis-related genes may further enhance the physiological functions of plants, such as antioxidant and stress resistance [22]. The MAPK signaling pathway is a key pathway for plants to transmit environmental stress signals and regulate stress-responsive responses, and its activation can regulate the expression of downstream stress-related genes through cascade reactions [23].

## Expression characteristics and pathway enrichment patterns of DMs under allelopathic stress

Metabolomic analysis confirmed that in *Elymus nutans* seedlings in response to stress, the changes of differential metabolites, metabolic pathways and gene expression were coordinated. DMs were significantly enriched in pathways such as secondary metabolite biosynthesis, ABC transporters, and amino acid biosynthesis. In the comparison between Group A and Group B, numerous related metabolic pathways including phenylpropanoid metabolism and α-linolenic acid metabolism were significantly activated. Among these, α-linolenic acid can be converted into jasmonic acid (JA), which further enhances stress signal transduction. Moreover, JA can jointly regulate the antioxidant system and secondary metabolic system, helping plants restore normal physiological status [24,25]. The enrichment of the ABC transporter pathway indicated that the transmembrane transport process of differential metabolites might be involved in the relevant regulatory mechanisms. ABC transporters can mediate the transmembrane transport of various metabolites and signaling molecules, and the activation of their related pathways ensures the precise intracellular distribution of differential metabolites [26].

## Synergistic effects of genes and metabolites under allelopathic stress

Integrated analysis revealed that under high-concentration stress, the obstruction of signal transduction pathways in seedlings led to a delayed activation of stress-resistant defense responses. Subsequently, the ribosome-mediated protein synthesis pathway was inhibited, resulting in insufficient supply of functional proteins. This further downregulated the secondary metabolic pathways and the cuticular barrier synthesis pathway, ultimately causing the loss of plant stress resistance. Moreover, the inhibition of the cuticular barrier synthesis pathway would further exacerbate the damage to both roots and above-ground tissues [27]. Under low-concentration treatment, seedlings regulate the degradation of ABA signaling factors and coordinate the rhythmic expression of antioxidant genes through the modulation of the circadian rhythm pathway, thus temporally coordinating seed germination and stress responses [28]. The core pathways in the A vs B groups included ribosome function, phenylpropanoid biosynthesis, and lipid metabolism. The activation of the ribosome pathway promoted the synthesis of functional proteins; lipid metabolism enhanced cell membrane integrity; and

the up-regulation of the phenylpropanoid pathway accelerated the production of defensive secondary metabolites. The synergy of multiple pathways enabled the molecular and metabolic recovery of plants.

### Regulatory mechanisms of phenylpropane metabolism under allelopathic stress

In this study, the significant inhibitory effect of MWA on *PAL* and *PTAL* at the initial stage of the pathway can reduce the supply of lignin synthesis precursors from the source by inhibiting the initial enzymes of the phenylpropane pathway [29], the inhibitory effect was more pronounced at high concentrations, enabling *Elymus nutans* to flexibly regulate the metabolic flux input according to stress intensity and thus avoid basal metabolic disorders caused by excessive pathway inhibition. Under low-concentration treatment, the weak regulatory effect on *HCT* inhibitory isoforms led to an increase in ferulic acid accumulation, whereas under high-concentration treatment, its strong inhibitory effect drove the conversion of ferulic acid to non-lignin branches. These results indicate that a small amount of ferulic acid accumulation under mild stress can balance cell wall structural stability and stress resistance demands, while under severe stress, metabolic flux redirection prioritizes the synthesis of stress-resistant active substances. This is consistent with the conclusions of previous studies that the accumulation of flavonoids and phenolic acids in plants under stress can improve antioxidant capacity and enhance stress tolerance [30], suggesting that MWA can balance the relationship between lignin synthesis and stress-resistant secondary metabolism by regulating *HCT* isoforms. As a key rate-limiting enzyme in cinnamyl alcohol synthesis, the continuous down-regulation of *CAD* under both low and high concentrations is the core step that blocks lignin monomer synthesis. In contrast, the activation of *CCR* at low concentrations and insufficient compensation at high concentrations reflects the compensatory mechanism of *Elymus nutans* in lignin synthesis regulation, which can avoid cell wall fragility caused by excessive lignin reduction under mild stress, and completely cut off the supply of lignin synthesis raw materials under severe stress [31]. The strong inhibition of peroxidase by MWA directly blocks the polymerization of lignin monomers, which, together with the inhibition of precursor synthesis, forms a dual-blockade mode. This dual regulatory pattern has also been reported in other stress-tolerant plants [32], indicating that it may represent a conserved strategy for lignin regulation in plant stress responses.

### Conclusion

This study clarified that MWA exerted a significant inhibitory effect on seed germination and seedling growth of *Elymus nutans*, which was specifically manifested by the obvious inhibition of seed GR and GE, as well as the average fresh weight and average root length of seedlings. Further studies revealed that *Elymus nutans* seedlings could enhance their resistance to allelopathic stress by regulating the expression levels of key genes including *PAL*, *PTAL*, *CAD*, *HCT* and *CCR*, as well as key metabolites such as p-coumaric acid, ferulic acid and sinapyl alcohol. Pathway analysis indicated that the biosynthesis of secondary metabolites, ribosome, plant MAPK signaling, and phenylpropanoid biosynthesis pathways were the common core pathways in response to different concentrations of MWA treatment. Among these, the phenylpropanoid biosynthesis pathway, as the core secondary metabolic pathway for *Elymus nutans* seedlings to respond to allelopathic stress, could mediate carbon flux allocation and signal transduction processes by regulating the synthesis of key metabolites such as lignin and flavonoids, synergistically enhancing the stress-resistant defense capability of seedlings. This study enriches the basic data resources of the transcriptome and metabolome of *Elymus nutans*, provides a theoretical basis for the excavation of key genes against allelopathic stress and the breeding of stress-resistant varieties, and also offers new insights for the vegetation restoration of degraded natural grasslands. Future studies could focus on the interaction mechanism between MWA and other signaling molecules, and further explore the universality of the aforementioned regulatory pathways under different types of stress, thereby laying a more solid theoretical foundation for the comprehensive elucidation of the stress-resistant adaptation mechanism of *Elymus nutans*.

## Supporting information

**S1 Data. Data of Fig 1.**
(XLSX)

**S2 Data. Data of Fig 2.**
(XLSX)

**S3 Data. Data of Fig 3.**
(XLSX)

**S4 Data. Data of Fig 4.**
(XLSX)

**S5 Data. Data of Fig 5.**
(XLSX)

**S6 Data. Data of Fig 6.**
(XLSX)

**S7 Data. Data of Fig 7.**
(XLSX)

**S8 Data. Data of Fig 8.**
(XLSX)

**S9 Data. Data of Fig 9.**
(XLSX)

## Acknowledgments

We sincerely thank Professor Yifu Wen for her guidance throughout this research and technical support from the project (Project No. 202502AE090016). We are grateful to the College of Chemistry, College of Agronomy, and College of Grassland Science at Gansu Agricultural University for providing laboratory space and equipment. We appreciate the assistance of Research Fellow Lianping Yu from Gansu Grassland Technology Extension Station and Professor Taotao Li from the College of Animal Science at Gansu Agricultural University in data analysis. We also thank Professor Lijing Yang from the College of Agronomy and Professor Kan Jiang from the College of Chemistry at Gansu Agricultural University for their help with experimental testing.

## Author contributions

**Data curation:** Huiyun Yu, Lianping Yu.

**Funding acquisition:** Huiyun Yu, Yifu Wen.

**Investigation:** Huiyun Yu, Lianping Yu.

**Methodology:** Jun Yin.

**Project administration:** Huiyun Yu, Xingming Liu.

**Software:** Lijing Yang.

**Supervision:** Yifu Wen.

**Validation:** Jie Zhang.

**Writing – original draft:** Huiyun Yu.

**Writing – review & editing:** Lianping Yu, Yifu Wen.

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
