## [Editor Report · Decision Letter 0]

31 Oct 2025

PONE-D-25-42682Elymus nutans Griseb responds to allelopathic stress of Ligularia sagitta through coordinated regulation of phenylpropanoid and flavonoid pathwaysPLOS ONE

Dear Dr. Wen,

Thank you for submitting your manuscript to PLOS ONE. After careful consideration, we feel that it has merit but does not fully meet PLOS ONE’s publication criteria as it currently stands. Therefore, we invite you to submit a revised version of the manuscript that addresses the points raised during the review process.

We look forward to receiving your revised manuscript.

Kind regards,

Chandran Janani

Academic Editor

PLOS ONE

Journal Requirements:

“This work was supported by the Gansu Provincial Forestry and Grassland Science and Technology Project (Project No. 2022kj074) and the Yunnan Plateau Characteristic Agriculture Science and Technology Program Project((Project No. 202502AE090016)”

**Additional Editor Comments:**

The manuscript presents a well-organized and comprehensive study integrating transcriptomic and metabolomic analyses to elucidate allelopathic stress responses in Elymus nutans. The work is scientifically sound and relevant to plant molecular biology and ecological physiology. However, the manuscript would benefit from improved clarity, conciseness, figure quality, and interpretation depth, especially in connecting omics data to physiological relevance.

Major Comments

• 1. Novelty and Scope: The integrated omics approach is commendable; however, similar pathway-level studies exist for allelopathy in other species. Clearly articulate the unique novelty of this work.

• 2. Experimental Design: Justify the choice of two concentrations (0.01 mg/mL and 0.10 mg/mL). Discuss ecological relevance and experimental rationale.

• 3. Data Integration: Present statistical correlations (e.g., Pearson's r) between DEGs and DAMs. Add a gene–metabolite correlation network or heatmap.

• 4. Figures: Simplify complex KEGG pathway figures and improve figure resolution and labeling consistency.

• 5. Biological Interpretation: Condense repetitive pathway descriptions and emphasize biological implications of metabolic remodeling.

• 6. Statistical Rigor: Specify biological vs. technical replicates and normalization methods used for RNA-seq and metabolomics data.

• 7. Language and Style: The manuscript requires grammatical polishing and editing for fluency and precision.

• 8. Conclusion: Focus more on ecological and physiological implications rather than repeating omics results.

Minor Comments

• 1. Include taxonomic verification and voucher specimen details.

• 2. Mention the number of biological replicates per treatment group.

• 3. Specify the transcriptome reference used for annotation.

• 4. Include qPCR primer sequences in supplementary data.

• 5. Define all abbreviations (e.g., CK, 2-A, 2-B) in figure legends.

• 6. Avoid redundant phrasing like 'significantly up/down-regulated'; use quantitative data.

• 7. Ensure consistent reference formatting and correct DOIs.

• 8. Verify all figure numbering and cross-references.

Recommendation

Decision: Major Revision

The study provides valuable insights but requires substantial revision in data presentation, figure quality, and interpretive clarity to meet publication standards.

---

## [Author Response · Author response to Decision Letter 1]

16 Jan 2026

Rebuttal Letter

Dear Editor,

Thank you very much for giving us the opportunity to revise our manuscript entitled “Transcriptomic and Metabolomic Analysis of Allelopathic Responses in Elymus nutans” (Manuscript ID: PONE-D-25-42682). We have revised the manuscript item by item in accordance with the comments provided by the editorial office.

Reviewers:

1.Journal Requirements

(2)In your Methods section, please provide additional information regarding the permits you obtained for the work.

(3)Please state what role the funders took in the study.

(4)Please confirm at this time whether or not your submission contains all raw data required to replicate the results of your study.Provide the original data related to this study

(5)Please ensure that you have an ORCID iD and that it is validated in Editorial Manager.

2.Additional Editor Comments

(1)the manuscript would benefit from improved clarity, conciseness, figure quality, and interpretation depth, especially in connecting omics data to physiological relevance.

3.Major Comments

(1)Novelty and Scope: The integrated omics approach is commendable; however, similar pathway-level studies exist for allelopathy in other species. Clearly articulate the unique novelty of this work.

(2) Experimental Design: Justify the choice of two concentrations (0.01 mg/mL and 0.10 mg/mL). Discuss ecological relevance and experimental rationale.

(3) Data Integration: Present statistical correlations between DEGs and DAMs. Add a gene–metabolite correlation network or heatmap.

(4)Figures: Simplify complex KEGG pathway figures and improve figure resolution and labeling consistency.

(5)Biological Interpretation: Condense repetitive pathway descriptions and emphasize biological implications of metabolic remodeling.

(6)Statistical Rigor: Specify biological vs. technical replicates and normalization methods used for RNA-seq and metabolomics data.

(7)Language and Style: The manuscript requires grammatical polishing and editing for fluency and precision.

(8)Conclusion: Focus more on ecological and physiological implications rather than repeating omics results.

4.Minor Comments

(1) Include taxonomic verification and voucher specimen details.

(2) Mention the number of biological replicates per treatment group.

(3)Specify the transcriptome reference used for annotation.

(4) Include qPCR primer sequences in supplementary data.

(5) Define all abbreviations (e.g., CK, 2-A, 2-B) in figure legends.

(6)Avoid redundant phrasing like 'significantly up/down-regulated'; use quantitative data.

(7) Ensure consistent reference formatting and correct DOIs.

(8) Verify all figure numbering and cross-references.

Response:

1.Journal Requirements

(1)The manuscript has been carefully checked in accordance with the PLOS ONE style templates, with all formatting errors revised.

(2)The statement of permission from the field sampling institution has been supplemented in the “Materials and Methods” section of the manuscript.The collection work has been authorized by the Luqu County Grassland Workstation of Gannan Prefecture. The plant specimens of Ligularia sagitta were authenticated by Professor Guo Yehong of Gansu Agricultural University, while those of Elymus nutans were identified by Researcher Yang Yandong of the Luqu County Grassland Workstation. All voucher specimens are deposited in Gansu Agricultural University.

(3)The statement on the role of the funding organization has been supplemented in the “Cover Cetter” and the “Author contributions” section of the manuscript.

(4)The raw sequence data reported in this paper have been deposited in the Genome Sequence Archive (Genomics,Proteomics & Bioinformatics 2025) in National Genomics Data Center (Nucleic Acids Res 2025),China National Center for Bioinformation / Beijing Institute of Genomics,Chinese Academy of Sciences (GSA: CRA033148) that are publicly accessible at https://ngdc.cncb.ac.cn/gsa/browse/CRA033148.

(5)The corresponding author has provided the ORCID ID in the Editorial Manager system,ORCID iD: 0009-0002-3069-7681.

2.Additional Editor Comments

(1)In accordance with the requirements of the editorial office, we have simplified the linguistic descriptions in the manuscript, redrawn all the figures and tables, and supplemented the correlation analysis between genes and metabolites.

3.Major Comments

(1)The significance and innovation of this study have been supplemented in Introduction of the manuscript.(At present, there is a distinct deficiency in the elucidation of molecular regulatory mechanisms underlying the allelopathic stress responses of recipient plants to Ligularia species. In particular, triterpenoids, which are ubiquitously present in plants, have been confirmed to exist in the roots of Ligularia sagitta and may be involved in allelopathic processes. However, their specific action targets, synergistic mechanisms with other terpenoids, and environmental adaptive response patterns remain unexplored research gaps to date. The findings of this study can break through the limitations of traditional single-component analysis, provide theoretical support for an in-depth understanding of the interactions between allelochemicals and plants, and meanwhile offer a scientific basis for the conservation and sustainable utilization of grassland ecosystems in alpine regions.)

(2)In the “Effects of different concentration treatments on seed germination and seedling growth of Elymus nutans”section of the Results, the reasons for selecting the two concentrations (0.01 mg/mL and 0.10 mg/mL) are illustrated by means of bar charts.

(3)In the “KEGG Pathway Enrichment and Correlation Analysis of DEMs and DMs”section of the Results,correlation coefficient matrix heatmaps and enrichment bubble plots of DEMs and DMs were supplemented for data integration analysis.

(4)In accordance with the experts’ comments, the GO functional annotation plot of differentially expressed genes and the bubble plot for KEGG pathway enrichment results in the “Screening and Functional Enrichment Analysis of DEMs”section of the Results in the manuscript were redrawn. In the “KEGG Pathway Enrichment and Correlation Analysis of DEMs and DMs”section of the Results, a correlation analysis heatmap matrix of genes and metabolites, a classification plot of KEGG pathway enrichment results, and a bubble plot of KEGG pathway enrichment were supplemented. In the “qRT-PCR Validation of DEGs”section of the Results, the linear fitting plot of qRT-PCR and RNA-seq results was redrawn. In the “Effects on the Biosynthesis of Phenylpropanoids”section of the Results, the plot of relative contents of key genes and main metabolites involved in the phenylpropane biosynthesis pathway was redrawn. In the “Effects of different concentration treatments on seed germination and seedling growth of Elymus nutans”section of the Results, the bar charts illustrating the effects of allelochemicals at different concentration gradients on the germination rate and germination energy of Elymus nutans seeds were redrawn, along with the bar charts depicting the inhibition rates of these allelochemicals on the shoot length and root length of Elymus nutans seedlings.

(5)In the “Effects on the Biosynthesis of Phenylpropanoids”section of the Results and the Discussion section, I focused on analyzing the metabolic pathway and products of the phenylpropanoid biosynthesis pathway. In response to exogenous stress, the synthesis of key metabolites (lignin and flavonoids) was regulated, which mediated carbon flux distribution and signal transduction processes, synergistically enhancing the stress resistance and defense capacity of Elymus nutans seedlings.

(6)In the “Experimental Materials and Treatments” section of Materials and Methods, details regarding the biological replicates and technical replicates for transcriptome and metabolome analyses have been supplemented. In the “Experimental Methods” section, specific normalization methods adopted for transcriptome and metabolome analyses have been added.

(7)I have carefully revised the manuscript, polished its grammar, improved its fluency, and refined the precision of the expressions.

(8)In accordance with the experts’ comments, in the Conclusion section, content beyond the scope of this study has been removed, along with repetitive and redundant statements. The ecological and physiological significance of this research has also been clarified.

4.Minor Comments

(1)In the "Experimental Materials and Treatments" section of Materials and Methods, details regarding taxonomic verification and voucher specimen have been supplemented.

(2)In the “Experimental Materials and Treatments” section of Materials and Methods, details regarding the biological replicates have been supplemented.

(3)Regarding the transcriptome reference used for annotation,since this study involved a non-model species without a reference genome, we performed de novo transcriptome sequencing (non-reference transcriptome sequencing). After sequencing,assembly was performed using Trinity(version: 2.11.0; parameters: -min_kmer_cov 2), and the longest transcript from each gene cluster was selected as a unigene. These assembled transcripts and unigenes were used as the reference sequences for subsequent gene annotation and expression analysis.

(4)In the “qRT-PCR Validation of DEGs” section within the Results of the manuscript, the qPCR primer sequences have been supplemented.

(5)All abbreviations included in the manuscript have been checked, and their definitions have been supplemented.

(6)I have carefully revised the manuscript to eliminate redundant phrasing such as "significantly up/down-regulated". Instead, we have replaced these descriptions with specific quantitative data corresponding to the up-regulated or down-regulated targets in all relevant sections of the manuscript.

(7)In accordance with the submission formatting guidelines of PLOS ONE, I have standardized the formatting of all references in the manuscript. Additionally, I have verified and corrected the DOIs of all references to ensure their accuracy and accessibility.

(8)I have thoroughly checked the numbering of all figures and tables in the manuscript to ensure sequential consistency. Following the extensive revisions made to the manuscript as requested by the editorial office, there are no cross-referenced references remaining in the manuscript.

---

## [Editor Report · Decision Letter 1]

27 Apr 2026

Transcriptomic and Metabolomic Analysis of Allelopathic Responses in Elymus nutans

PONE-D-25-42682R1

Dear Dr. Wen,

We’re pleased to inform you that your manuscript has been judged scientifically suitable for publication and will be formally accepted for publication once it meets all outstanding technical requirements.

Kind regards,

Chandran Janani

Academic Editor

PLOS One

Additional Editor Comments (optional):

Accept the manuscript with this revised one
---

## [Editor Report · Acceptance letter]

PONE-D-25-42682R1

PLOS One

Dear Dr. Wen,

I'm pleased to inform you that your manuscript has been deemed suitable for publication in PLOS One. Congratulations! Your manuscript is now being handed over to our production team.

Kind regards,

on behalf of

Dr. Chandran Janani

Academic Editor

PLOS One